# On learning sparse vectors from mixture of responses

**Nikita Polyanskii**
IOTA Foundation
Berlin, Germany
nikitapolyansky@gmail.com

## Abstract

In this paper, we address two learning problems. Suppose a family of $\ell$ unknown sparse vectors is fixed, where each vector has at most $k$ non-zero elements. In the first problem, we concentrate on robust learning the supports of all vectors from the family using a sequence of noisy responses. Each response to a query vector shows the sign of the inner product between a randomly chosen vector from the family and the query vector. In the second problem, we aim at designing queries such that all sparse vectors from the family can be approximately reconstructed based on the error-free responses. This learning model was introduced in the work of Gandikota et al., 2020, and these problems can be seen as generalizations of support recovery and approximate recovery problems, well-studied under the framework of 1-bit compressed sensing. As the main contribution of the paper, we prove the existence of learning algorithms for the first problem which work without any assumptions. Under a mild structural assumption on the unknown vectors, we also show the existence of learning algorithms for the second problem and rigorously analyze their query complexity.

## 1 Introduction

Many digital data acquisition methods can be expressed as recovering an unknown real-valued sparse vector from linear measurements. Compressed sensing is a signal processing methodology proposed in [9, 5] to make the process of vector reconstruction reliable and efficient. Criticism of classical compressed sensing is that it assumes infinite precision of real-valued measurements. One interesting setup for compressed sensing is that the measurement outcomes are quantized, e.g., only signs of measurements are captured. Different aspects of the 1-bit compressed sensing problem have been extensively investigated in recent years [1, 3, 25, 2, 16, 18, 13]. One important sub-problem that often arises in 1-bit compressed sensing is to learn the support of an unknown sparse vector. A vast body of literature focuses on this task and gives bounds on the minimum number of measurements sufficient for support recovery [1, 16, 13].

In this paper, we investigate a model which can be seen as a generalization of 1-bit compressed sensing. In this model, a family of $\ell$ unknown sparse $k$-sparse vectors in $\mathbb{R}^n$ is fixed, and the response to a query vector shows the sign of the inner product between the query vector and a vector picked uniformly at random from the family. The goal is to reconstruct the vectors (or only the supports of the vectors) using the least number of queries. In real-life applications, the measurements are suffered by noise, which causes sign flips. Thereby, we also suggest considering a variation of the model that admits adversarial noise.

Our problem statement is in spirit of many machine learning models, e.g., a mixture of linear regression models [7, 19, 27, 21, 17, 30, 29, 6] and a mixture of binary linear classifiers [29]. However, we stress an important difference from the classical machine learning setting: in our model, one constructs data points and an oracle provides the label for each of them; the goal is to find the minimal number of data points sufficient for learning all the components in a mixture of binary linear

35th Conference on Neural Information Processing Systems (NeurIPS 2021).

classifiers. Recent works on a mixture of sparse linear regressions considered a similar setting and studied query complexity of learning algorithms [31, 23, 20].

The most relevant research to our work is provided in [14], where the noiseless version of the proposed model was discussed and some single-stage and two-stage learning strategies were provided. The main technical contribution presented in that paper is an algorithm for finding the support of the sparse vectors. However, its main drawback is that for three or more unknown vectors, it was shown to work only under the support separability assumption: the support of any unknown vector in the family is not covered by the union of the supports of the other unknown vectors (c.f. Table 1). The authors left the problem of support recovery without any structural assumptions as an open problem. We also highlight that for a family of two unknown sparse vectors, the authors of [14] provided a polynomial-time algorithm for the approximate reconstruction of both vectors that works under a mild assumption.

Since the first submission of this work, the authors of [15] have developed additional theoretical results for the robust support recovery problem. Specifically, using various tensor decomposition techniques and constructions of union-free families, it was shown how to efficiently learn with high probability the supports of all unknown vectors without any assumptions (c.f. Table 1). For certain structural assumptions on the support of unknown vectors, it was also shown how to improve the query complexity.

## 1.1 Our contribution and technique

Our main contribution lies in showing the existence of a learning algorithm that works without any assumptions and is resilient to noisy measurements. Suppose a family of $\ell$ unknown $k$-sparse vectors in $\mathbb{R}^n$ is fixed. For any given subset $Q \subset [n]$, we first design queries that enable us to learn the number of vectors whose support intersects $Q$ non-trivially. Based on this knowledge, we show how to reduce the support recovery problem to a problem of quantitative group testing. In this problem, one needs to learn a hidden hypergraph defined over the set $[n]$ with $\ell$ edges each of size at most $k$. The response to a query $Q \subset [n]$ shows the piercing number, the number of edges overlapping with $Q$. Here, we arrive to the concept of $(\ell, k, \alpha)$-robust-resolvable matrices (Definition 1) that allow us to uniquely reconstruct any hidden sparse hypergraph based on a sequence of responses. In the combinatorial coding theory literature, this concept was analyzed only when $k = 1$, $\alpha = 0$ [26]. We extend this line of research for $k \geq 1$ and $\alpha \geq 0$ and give an achievability bound for robust-resolvable matrices. In particular, we show a connection between robust-resolvable matrices and a notion that generalizes cover-free families which were previously investigated in extremal set theory [28, 12]. We highlight that the use of some specific subclasses of generalized cover-free families such as union-free families and pairwise independent union-free families was already discussed in the context of various 1-bit compressed sensing problems [14, 1, 16]. However, such a link was used before either in a very specific setting, e.g., $\ell = 1$, or under certain assumptions. We note that all queries of the proposed support recovery algorithm can be carried out in parallel (Theorem 1).

Once the supports of all unknown vectors in the family are reconstructed, it remains to recover the values of these vectors. We consider this task in the noiseless setting and make an assumption: (a) the support of any unknown vector is not fully contained in the support of any other unknown vector from the family, and (b) the magnitude of each entry can be bounded from below and above. It is known that $\mathcal{O}(k/\epsilon \log(n/\epsilon))$ simple Gaussian queries are sufficient to learn an unknown $k$-sparse vector in $\mathbb{R}^n$ using only signs of linear measurements [18]. Thereby, one natural idea to recover the family of vectors is to model a Gaussian query for every given unknown vector from the family. A similar technique was also used in [14], however, our solutions work under a milder assumption. In a simple two-stage recovery process, we carry out queries required for the approximate reconstruction at the second stage after getting responses to queries used for the support recovery (Theorem 2). In a more sophisticated non-adaptive solution, we re-use the concept of cover-free families that enable us to emulate individual Gaussian queries for any potential family of $\ell$ unknown sparse vectors (Theorem 3).

We emphasize that our contribution is primarily theoretical as the questions raised in our work are purely mathematical. For instance, we leave the problem of designing a query scheme that allows for an efficient reconstruction algorithm with running time polynomial in the number of queries as an open problem. However, our research has relevant practical applications since the considered model can be used for modeling complex systems with heterogeneous data. According to [14], studying

different aspects in such a model would help analyze recommendation systems where the privacy of user's data has to be preserved.

## 1.2 Outline

The remainder of the paper is organized as follows. We introduce some notations and describe the problem statement in Sec. 2. Our results are summarized in Sec. 3. Sec. 4 describes a robust support recovery algorithm and introduces useful concepts required for the algorithm. We present our approximate recovery algorithms in Sec. 5. Finally, Sec. 6 concludes the paper and discusses some open research directions. The proofs of many statements are deferred to the supplementary material.

## 2 Problem statement

For simplicity of presentation, hereafter one-based numbering is used and $\log n$ stands for the base-two logarithm of $n$. Let $H_2(\alpha)$ denote the binary entropy function, i.e., $H_2(\alpha) = -\alpha \log \alpha - (1 - \alpha) \log(1 - \alpha)$. The set of integers from 1 to $n$ is abbreviated by $[n]$. A vector is denoted by bold lowercase letters, such as $\boldsymbol{x}$, and the $i$-th entry of the vector $\boldsymbol{x}$ is referred to as $x_i$. A binary matrix is denoted by uppercase letters, such as $X$, and the entry in the $i$-th row and the $j$-th column is written as $X_{i,j}$.

We say that a vector $\boldsymbol{x} \in \mathbb{R}^n$ is $k$-*sparse* if it contains at most $k$ non-zero components. The *support* of a vector $\boldsymbol{x}$, written as $\mathrm{supp}(\boldsymbol{x})$, denotes the set of coordinates corresponding to non-zero components of $\boldsymbol{x}$. We write $\mathbb{1}\{A\}$ to denote the indicator function of an event $A$. For an integer vector $\boldsymbol{x} \in \mathbb{Z}^m$ and an integer $r$, $0 \le r \le m$, define the *Hamming ball* centered at $\boldsymbol{x}$ with radius $r$ as $B_r(\boldsymbol{x}) := \{\boldsymbol{y} \in \mathbb{Z}^m : d_H(\boldsymbol{x}, \boldsymbol{y}) \le r\}$, where $d_H(\boldsymbol{x}, \boldsymbol{y})$ denotes the Hamming distance between $\boldsymbol{x}$ and $\boldsymbol{y}$. We will use the standard notion of $\ell^2$-norm written as $\|\cdot\|_2$.

Let $\mathcal{B} = \{\boldsymbol{\beta}^{(1)}, \ldots, \boldsymbol{\beta}^{(\ell)}\}$ be a family of $\ell$ unknown $k$-sparse vectors in $\mathbb{R}^n$ such that $\left\|\boldsymbol{\beta}^{(i)}\right\|_2 = 1$ for all $i \in [\ell]$. Throughout the paper we assume that the number of vectors is not too large and the vectors are indeed sparse. Specifically, let $\ell k = o(n)$. Suppose that an oracle returns a (noisy) output $y_i \in \{-1, 1\}$ for a given query vector $\boldsymbol{x}^{(i)} \in \mathbb{R}^n$, $i \in [q]$:

$$y_i = \mathrm{sign}(\langle \boldsymbol{x}^{(i)}, \boldsymbol{\beta} \rangle) \cdot \eta_i, \tag{1}$$

where the random variable $\boldsymbol{\beta}$ is sampled uniformly at random from the set $\mathcal{B}$, the value $\eta_i \in \{-1, 1\}$ is noise, and the sign function is defined as follows

$$\mathrm{sign}(\alpha) = \begin{cases} 1, & \alpha \ge 0, \\ -1, & \alpha < 0. \end{cases}$$

Clearly, for $\eta_i = 1$, the resulting response is exactly the sign of the inner product between $\boldsymbol{x}^{(i)}$ and $\boldsymbol{\beta}$. For the approximate recovery problem, we assume the noiseless case, i.e., $\eta_i = 1$ for all $i \in [q]$. For the support recovery problem, we make the only structural assumption regarding the noise level, namely:

$$\sum_{i=1}^{q} \mathbb{1}\{\eta_i = -1\} \le \tau q \tag{2}$$

for a given real value $\tau \in [0, 1]$. Our first goal is to recover the supports of all unknown vectors.

**Problem 1** ($\tau$-robust support recovery)**.** *Let $\tau$ be a real value from the interval $[0, 1]$ and $\mathcal{B} = \{\boldsymbol{\beta}^{(1)}, \ldots, \boldsymbol{\beta}^{(\ell)}\}$ be a family of $\ell$ unknown $k$-sparse vectors in $\mathbb{R}^n$. Design queries $\boldsymbol{x}^{(1)}, \ldots, \boldsymbol{x}^{(q)}$ such that with probability $1 - \mathcal{O}(1/n)$, based on a sequence of noisy binary responses $y_1, \ldots, y_q$, provided that (1)-(2), it is possible to find $S_1, \ldots, S_\ell \subseteq [n]$ with*

$$\{S_1, \ldots, S_\ell\} = \{\mathrm{supp}(\boldsymbol{\beta}^{(1)}), \ldots, \mathrm{supp}(\boldsymbol{\beta}^{(\ell)})\}.$$

Our second task is to approximately recover the unordered set of directions of $\ell$ unknown vectors with a given precision in the noiseless setting. Formally, this problem can be stated as follows.

| Result | Query complexity | Assumption |
|---|---|---|
| [14, Theorem 1] | $\mathcal{O}(\ell^6 k^3 \log^2 n)$ | the support of any unknown vector is not covered by the union of the supports of other vectors |
| [15, Corollary 1] | $\mathcal{O}(\ell^3 (\ell k)^{\log \ell + 2} \log^2 n)$ | No |
| Theorem 1 | $\mathcal{O}(k\ell^3 \max(7\ell, 7k)^{\min(\ell, k)} \log^2 n)$ | No |

Table 1: Comparison of query complexity for support recovery of $\ell$ unknown $k$-sparse vectors in $\mathbb{R}^n$.

**Problem 2** ($\epsilon$-approximate recovery)**.** *Fix $\tau = 0$. Let $\epsilon > 0$ be a given real value and $\mathcal{B} = \{\boldsymbol{\beta}^{(1)}, \ldots, \boldsymbol{\beta}^{(\ell)}\}$ be a family of $\ell$ unknown $k$-sparse vectors in $\mathbb{R}^n$. Design queries $\boldsymbol{x}^{(1)}, \ldots, \boldsymbol{x}^{(q)}$ such that with probability $1 - \mathcal{O}(1/n)$, based on the sequence of binary responses $y_1, \ldots, y_q$ provided that (1)-(2), it is possible to reconstruct the unknown vectors with precision $\epsilon$, i.e., find $k$-sparse vectors $\tilde{\boldsymbol{\beta}}^{(1)}, \ldots, \tilde{\boldsymbol{\beta}}^{(\ell)} \in \mathbb{R}^n$ satisfying the inequality*

$$\max_{i \in [\ell]} \left\| \boldsymbol{\beta}^{(i)} - \tilde{\boldsymbol{\beta}}^{(\sigma(i))} \right\|_2 \leq \epsilon,$$

*where $\sigma : [\ell] \to [\ell]$ is some bijection (permutation).*

## 3   Our results

In the following three statements, we summarize our main results concerning the above two problems. First, we show the existence of a robust support recovery algorithm without any assumptions.

**Theorem 1** (Robust support recovery algorithm)**.** *Let $\mathcal{B} = \{\boldsymbol{\beta}^{(1)}, \ldots, \boldsymbol{\beta}^{(\ell)}\}$ be a family of $\ell$ unknown $k$-sparse vectors in $\mathbb{R}^n$. Let $\tau \in [0, \frac{p'}{32\ell})$, where*

$$p' := \left( \frac{\min(\ell, k)}{2k\ell} \right)^{\min(\ell, k)} \left( 1 - \frac{\min(\ell, k)}{2k\ell} \right)^{2k\ell - \min(\ell, k)}. \tag{3}$$

*Define*

$$w(\gamma, \ell, k) := (2\gamma - 1)\log(1 - p') - 2\gamma \log p' - H_2(2\gamma).$$

*Then there exists a $\tau$-robust algorithm to learn the support of all vectors in $\mathcal{B}$ with probability at least $1 - \mathcal{O}(1/n)$ using*

$$\frac{(2k\ell - 1) \log n}{w(16\tau\ell, \ell, k)} (1 + o(1))$$

*non-adaptive queries, where $e$ is Euler's number. For the noiseless case $\tau = 0$, the query complexity can be estimated as*

$$\mathcal{O}(k\ell^3 \max(2e\ell, 2ek)^{\min(\ell, k)} \log^2 n).$$

**Remark 1.** *For the noiseless case $\tau = 0$, we compare our query complexity for the support recovery problem with the results of prior work [14] and parallel paper [15] in Table 1. Similar to [15] and unlike [14], we don't impose any structural assumptions on the unknown vectors, which may make the learning problem much simpler. The main drawbacks of our recovery algorithm are: (i) the running time is exponential in $k$ and $\ell$, (ii) the query complexity is exponential in $\min(k, \ell) \log(\max(k, \ell))$. Our solution outperforms the one from [15] for small enough $k$, say $k = o(\log \ell)$. We believe that the dependence on $k$ and $\ell$ is not optimal in all existing solutions and it is possible to further improve the query complexity. For instance, for the degenerate case $k = 1$ and $\tau = 0$, we also show the existence of an algorithm with $\mathcal{O}(\frac{\ell \log^2 n}{\log \ell})$ queries. In our proof, we think that the employed construction of robust-resolvable matrices via robust cover-free codes is far from an optimal one (Lemma 5). As for converse bounds in the noiseless setting, from simple counting arguments, it follows that the number of queries in the support recovery problem has to be at least $\Omega(\frac{k\ell}{\log(\ell+1)} \log n)$. Recall that for $\ell = 1$, the number of queries required for finding the support of the unknown vector is at least $\Omega(\frac{k^2}{\log k} \log n)$ which is implied by [11, 1].*

Let us introduce a mild assumption required for the approximate recovery problem.

**Assumption 1.** *Let $\mathcal{B} = \{\boldsymbol{\beta}^{(1)}, \ldots, \boldsymbol{\beta}^{(\ell)}\}$ be a family of $\ell$ unknown $k$-sparse vectors in $\mathbb{R}^n$. For every $i \neq j$, the support of the $i$-th vector from $\mathcal{B}$ is not contained in the support of the $j$-th vector from $\mathcal{B}$, i.e., $\mathrm{supp}(\boldsymbol{\beta}^{(i)}) \setminus \mathrm{supp}(\boldsymbol{\beta}^{(j)}) \neq \emptyset$. Some bounds on the magnitude of non-zero entries in all vectors from $\mathcal{B}$ are given, e.g., $\exp(-\mathcal{O}(n)) \leq |\beta_j^{(i)}| \leq \exp(\mathcal{O}(n))$ for all $j \in [n]$ and $i \in [\ell]$ such that $\beta_j^{(i)} \neq 0$.*

Recall that the approximate recovery problem is considered in the noiseless setting, i.e., $\tau = 0$. Now we show the existence of a query algorithm which can be performed in two (non-adaptive) stages.

**Theorem 2** (Two-stage approximate recovery algorithm). *Let $\epsilon > 0$ be a real value. Let $\mathcal{B} = \{\boldsymbol{\beta}^{(1)}, \ldots, \boldsymbol{\beta}^{(\ell)}\}$ be a family of vectors as in Assumption 1. Then there exists an algorithm to recover all vectors in $\mathcal{B}$ with precision $\epsilon$ with probability at least $1 - \mathcal{O}(1/n)$ using*

$$\mathcal{O}\left( k\ell^3 \max(2e\ell, 2ek)^{\min(\ell,k)} \log^2 n + \frac{k\ell^3}{\epsilon} \log(k/\epsilon) \log(n/\epsilon) \right)$$

*queries that can be asked in two (non-adaptive) stages.*

Finally, we present a result concerning the existence of non-adaptive learning scheme.

**Theorem 3** (Single-stage approximate recovery algorithm). *Under the condition of Theorem 2, there exists an algorithm to recover all vectors in $\mathcal{B}$ with precision $\epsilon$ with probability at least $1 - \mathcal{O}(1/n)$ using*

$$\mathcal{O}\left( \frac{k^2\ell^3}{\epsilon} \max(2e\ell, 2ek)^{\min(k,\ell)} \log n \log(n/\epsilon) \log(k/\epsilon) \right)$$

*non-adaptive queries.*

## 4  Robust support recovery

For a set $Q \subseteq [n]$ and a family $\mathcal{B} = \{\boldsymbol{\beta}^{(1)}, \ldots, \boldsymbol{\beta}^{(\ell)}\}$ of $\ell$ unknown $k$-sparse vectors in $\mathbb{R}^n$, we define the *piercing number* of $Q$ in $\mathcal{B}$ as

$$\mathrm{pir}(Q, \mathcal{B}) := \sum_{i=1}^{\ell} \mathbb{1}\{\mathrm{supp}(\boldsymbol{\beta}^{(i)}) \cap Q \neq \emptyset\}.$$

In Sec. 4.1 we first show how to learn this number with overwhelming probability by asking the same query multiple times. In Sec. 4.2 we rephrase the support recovery problem in the language of quantitative group testing and introduce the concept of robust resolvable matrices. The existence of such matrices is discussed in Sec. 4.3 and Sec. 4.4. Finally, in Sec. 4.5 we give a proof of Theorem 1.

### 4.1  Estimating the piercing number

Let $Q \subseteq [n]$ and $\boldsymbol{x} = (x_1, \ldots, x_n)$ be a vector such that the $i$-th entry is sampled independently from the continuous uniform distribution $U(0, 1)$ if $i \in Q$ and is 0 otherwise. For an integer $m$ (divisible by $16\ell$), define

$$\boldsymbol{x}^{(1)} = \ldots = \boldsymbol{x}^{(m/2)} = \boldsymbol{x}$$

and

$$\boldsymbol{x}^{(m/2+1)} = \ldots = \boldsymbol{x}^{(m)} = -\boldsymbol{x}.$$

For every $i \in [m]$, let $y_i$ be a noisy response defined as in (1). Let $\mathrm{round}(\gamma)$ be a function that takes a real number $\gamma$ as input and outputs the closest integer number. Define a sample statistic that estimates $\mathrm{pir}(Q, \mathcal{B})$

$$\widetilde{\mathrm{pir}}(Q, m) := \mathrm{round}\left( \frac{\ell}{m} \left( \sum_{i=1}^{m/2} y_i - \sum_{i=m/2+1}^{m} y_i \right) \right).$$

**Lemma 1** (Piercing number estimator). *Suppose $\sum_{i=1}^{m} \mathbb{1}\{\eta_i = -1\} \leq \frac{m}{16\ell}$. Then it holds that*

$$\Pr\left\{ \widetilde{\mathrm{pir}}(Q, m) = \mathrm{pir}(Q, \mathcal{B}) \right\} = 1 - \exp(-\Omega(m\ell^{-2})).$$

## 4.2 Quantitative search model for sparse hypergraphs

A hypergraph is a pair $H = (V, E)$, where $V$ is a set of vertices, and $E$ is a multiset of non-empty subsets of $V$ called hyperedges. A hypergraph $H = (V, E)$ is said to be $(\ell, k)$-*sparse* if the number of hyperedges is $\ell$ and the size of each hyperedge is at most $k$. Let $E = \{e_1, \dots, e_\ell\}$. For a subset $Q \subseteq V$, define the integer output $z(Q, H)$ as follows

$$z(Q, H) := \sum_{i=1}^{\ell} \mathbb{1} \{e_i \cap Q \neq \emptyset\}. \tag{4}$$

We now introduce the hypergraph $H_{\mathcal{B}} := (V, E_{\mathcal{B}})$, where the set of vertices $V = [n]$ and the set of hyperedges $E_{\mathcal{B}} = \{e_1, \dots, e_\ell\}$ contains $\ell$ hyperedges with $e_i = \mathrm{supp}(\boldsymbol{\beta}^{(i)})$ for all $i \in [\ell]$. Note that without any assumptions, the edge set $E_{\mathcal{B}}$ can possibly contain a hyperedge with large multiplicity if some vectors in $\mathcal{B}$ share the same support. Clearly, $H_{\mathcal{B}}$ is $(\ell, k)$-sparse and $z(Q, H_{\mathcal{B}}) = \mathrm{pir}(Q, \mathcal{B})$. In next statement, we derive an obvious extension of Lemma 1.

**Lemma 2.** *Let $\mathcal{B}$ be a family of unknown sparse vectors and $Q_1, \dots, Q_m \subseteq [n]$ be a sequence of subsets. Set $\boldsymbol{z} = (z(Q_1, H_{\mathcal{B}}), \dots, z(Q_m, H_{\mathcal{B}}))$. Let $\tau$ be a noise parameter such that $0 \leq \tau \leq \frac{1}{16\ell}$. Then there exist queries $\boldsymbol{x}^{(1)}, \dots, \boldsymbol{x}^{(q)} \in \mathbb{R}^n$ with $q = \mathcal{O}(\ell^2 m \log(mn))$ such that with probability $1 - \mathcal{O}(1/n)$, based on a sequence of binary responses $y_1, \dots, y_q$ provided that (1)-(2), it is possible to reconstruct at least one $\tilde{\boldsymbol{z}} \in \mathbb{Z}^m$ such that $d_H(\tilde{\boldsymbol{z}}, \boldsymbol{z}) \leq 16\tau\ell m$.*

Let $X$ be a binary matrix of size $m \times n$, and $H = (V, E)$ be a hypergraph with $V = [n]$. Set $Q_i = \mathrm{supp}(X_i)$, where $X_i$ is the $i$-th row of $X$. With a slight abuse of notation, we denote by

$$\boldsymbol{z}(X, H) := (z_1, \dots, z_m)$$

the sequence of integer responses, where $z_i = z(Q_i, H)$ is computed as in (4). Lemma 2 naturally motivates us to introduce a notion of robust resolvable matrices.

**Definition 1** (Robust resolvable matrix)**.** Fix a real number $\alpha, 0 \leq \alpha < 1/2$, and integers $\ell, k \geq 1$. A matrix $X$ of size $m \times n$ is said to be $(\ell, k, \alpha)$-*robust-resolvable* if for any two distinct $(\ell, k)$-sparse hypergraphs $H_1, H_2$, the corresponding Hamming balls centered at $\boldsymbol{z}(X, H_1)$ and $\boldsymbol{z}(X, H_2)$ with radius $\alpha m$ are disjoint, i.e.,

$$d_H(\boldsymbol{z}(X, H_1), \boldsymbol{z}(X, H_2)) \geq 2\alpha m + 1.$$

By $m(n, \ell, k, \alpha)$ denote the minimal number of rows in an $(\ell, k, \alpha)$-robust-resolvable matrix with $n$ columns. Up to our best knowledge, Definition 1 for $\alpha > 0$ (and $\alpha = 0$, $k \geq 2$) is new in the literature. We proceed our discussion with the simplest case $k = 1$ which might seem degenerate. However, the corresponding analysis is already non-trivial and sheds light on the existence of robust resolvable matrices.

## 4.3 Existence of $(\ell, 1, 0)$-robust-resolvable matrices

We now review literature on $(\ell, 1, 0)$-resolvable matrices. The best known explicit constructions come from number theory [4, 22] and are closely related to the concept of BCH codes. They have $n$ columns and $m = \ell \log n(1 + o(1))$ rows. For $\ell \geq 3$, the best currently known resolvable matrices are all inexplicit and constructed by random coding as shown in [26, 10].

**Lemma 3.** *For any given integer $\ell$, there exists an $(\ell, 1, 0)$-resolvable matrix of size $m \times n$ with*

$$m = \frac{2\ell - 1}{\log \binom{2^{2\ell}}{\binom{2\ell}{\ell}}} \log n(1 + o(1)).$$

*Furthermore, $m = 4\ell \log n / \log \ell (1 + o(1))$ as $\ell \to \infty$.*

**Remark 2.** *The presented existential result is order-optimal. By [10, 8], the number of rows in an $(\ell, 1, 0)$-robust-resolvable matrix with $n$ columns has to be $m \geq 2\ell \log n / \log \ell (1 + o(1))$.*

## 4.4 Constructing robust resolvable matrices via robust cover-free matrices

It is challenging to find a sufficient (and necessary) condition for a matrix to be $(\ell, k, \alpha)$-robust-resolvable. In the following statement, we make one step toward understanding this problem.

**Lemma 4.** *Suppose $H_1 = (V, E_1)$ and $H_2 = (V, E_2)$ are two different $(\ell, k)$-sparse hypergraphs. Then there exists a query $Q \subseteq V$ of size at most $\min(\ell, k)$ that separates the hypergraphs, i.e., $z(Q, H_1) \neq z(Q, H_2)$.*

*Proof.* Since two hypergraphs are distinct, there exists at least one hyperedge such that one edge set ($E_1$ or $E_2$) contains it with a larger multiplicity than the other one ($E_2$ or $E_1$). We take the largest hyperedge with this property. Without loss of generality, let $m_1$ and $m_2$ be multiplicities such that $e$ is included, respectively, to $E_1$ with multiplicity $m_1$ and to $E_2$ with multiplicity $m_2$, $\ell \geq m_1 > m_2 \geq 0$. For a vertex $v$, define the set of hyperedges in a hypergraph $H = (V, E)$ that are incident to this vertex

$$A(v, H) := \{e' \in E : v \in e'\}.$$

Consider the set of hyperedges $\bigcap_{v \in e} A(v, H_1)$. It includes $m_1$ copies of $e$ and possibly some other hyperedges $E' \subseteq E_1$. We observe that the set of hyperedges $\bigcap_{v \in e} A(v, H_2)$ includes $m_2$ copies of $e$ and precisely the same set of hyperedges $E'$. This happens due to the choice of $e$. Thus, we obtain

$$m_1 + |E'| = \left| \bigcap_{v \in e} A(v, H_1) \right| > \left| \bigcap_{v \in e} A(v, H_2) \right| = m_2 + |E'|. \tag{5}$$

Toward a contradiction, assume that there is no $Q \subseteq e$ such that

$$z(Q, H_1) = \left| \bigcup_{v \in Q} A(v, H_1) \right| \neq \left| \bigcup_{v \in Q} A(v, H_2) \right| = z(Q, H_2).$$

This means that the above unions for any $Q \subseteq e$ must have the same size. By the inclusion-exclusion principle

$$\left| \bigcup_{v \in Q} A(v, H_1) \right| = \sum_{\emptyset \neq Q' \subseteq Q} (-1)^{|Q'|+1} \left| \bigcap_{v \in Q} A(v, H_1) \right|.$$

Therefore, by induction on the size of $Q$, one can derive

$$\left| \bigcap_{v \in Q} A(v, H_1) \right| = \left| \bigcap_{v \in Q} A(v, H_2) \right|$$

for all $Q \subseteq e$. This contradicts to (5) when $Q = e$.

The above arguments show that a proper query $Q$ could have size at most $\ell$. Assume that $z(Q, H_1) > z(Q, H_2)$. Note that $z(Q, H_1) \leq k$ since $H_1$ contains $k$ edges. Thus, one can remove at least $\max(0, |Q| - k)$ vertices from $Q$ to get a subset of vertices $Q'$ such that the output result for $H_1$ is not changed, i.e., $z(Q', H_1) = z(Q, H_1)$. On the other hand, it is clear that $z(Q', H_2) \leq z(Q, H_2)$. This completes the proof. $\square$

Lemma 4 suggests a way for constructing $(\ell, k, \alpha)$-robust-resolvable matrices. We now introduce a concept that generalizes the definition of cover-free codes. This family of codes was originally suggested in [24] in connection with cryptographic applications.

**Definition 2.** A binary matrix $X$ of size $m \times n$ is said to be $(c, f, \alpha)$-robust-cover-free if for any disjoint subsets of columns $U, W \subseteq [n]$ of size $|U| = c$ and $|W| = f$, there exists at least $2\alpha m + 1$ rows $I \subseteq [m]$ such that

$$X_{i,j} = 1 \quad \text{for all } i \in [I], j \in U, \qquad X_{i,j} = 0 \quad \text{for all } i \in [I], j \in W.$$

If $\alpha = 0$, then a $(c, f, \alpha)$-robust-cover-free matrix is simply called $(c, f)$-cover-free.

An $(1, f, \alpha)$-robust-cover-free matrix is also called robust union-free. The latter has several applications in 1-bit compressed sensing [1, 16]. Now we state a straightforward corollary of Lemma 4 that enables us to construct robust resolvable matrices.

**Lemma 5.** *Suppose a binary matrix $X$ of size $m \times n$ is $(\min(\ell, k), 2k\ell - \min(\ell, k), \alpha)$-robust-cover-free. Then $X$ is $(\ell, k, \alpha)$-robust-resolvable.*

*Proof.* Let $H_1 = (V, E_1)$ and $H_2 = (V, E_2)$ be two distinct $(\ell, k)$-sparse hypergraphs. By Lemma 4, there exists a query $Q \subseteq V$, $|Q| \leq \min(\ell, k)$, such that $z(Q, H_1) \neq z(Q, H_2)$. Define $W$ to be $\{v \in e : e \in E_1 \cup E_2\}$. Since we associate $V$ with $[n]$, we get $Q \subseteq [n]$, $|Q| \leq \min(\ell, k)$, and $W \subseteq [n]$, $|W| \leq 2\ell k - \min(\ell, k)$. Combining these arguments with the fact that $X$ is $(\min(\ell, k), 2k\ell - \min(\ell, k), \alpha)$-robust-cover-free yields that $X$ is $(\ell, k, \alpha)$-robust-resolvable. □

Fundamental limits of cover-free matrices have been a subject of extensive studies in many papers [12, 28]. However, we cannot directly apply known bounds since they correspond to the case $\alpha = 0$. Now we present an achievability bound for robust cover-free matrices.

**Lemma 6.** *Let $c \geq 1$ and $f \geq 1$ be integers, $\alpha$ be a real number from the interval $[0, \overline{p}/2)$, where*

$$\overline{p} := \left(\frac{c}{c+f}\right)^c \left(\frac{f}{c+f}\right)^f.$$

*Define*

$$g(\gamma, c, f) := (2\gamma - 1)\log(1 - \overline{p}) - 2\gamma \log \overline{p} - H_2(2\gamma).$$

*Then there exists a $(c, f, \alpha)$-robust-cover-free matrix of size $m \times n$ such that*

$$m \leq \frac{(c + f - 1)\log n}{g(\alpha, c, f)}(1 + o(1)).$$

Using the above statement, we show an existential result for robust-resolvable matrices.

**Lemma 7.** *Let $\ell \geq 1$ and $k \geq 1$ be integers, $\alpha$ be a real number from the interval $[0, p'/2)$. Let $p'$ and $w(\gamma, \ell, k)$ be as in Theorem 1. Then there exists an $(\ell, k, \alpha)$-robust-resolvable matrix of size $m \times n$ such that*

$$m \leq \frac{(2k\ell - 1)\log n}{w(\alpha, \ell, k)}.$$

*For $\alpha = 0$, we have*

$$m(n, \ell, k, \alpha) = \mathcal{O}(k\ell \max(2e\ell, 2ek)^{\min(\ell, k)}\log n).$$

*Proof.* The statement is implied by Lemma 5 and Lemma 6 with $c = \min(\ell, k)$, $f = 2\ell k - \min(\ell, k)$. □

### 4.5 Proof of Theorem 1

Let $\mathcal{B} = \{\boldsymbol{\beta}^{(1)}, \ldots, \boldsymbol{\beta}^{(\ell)}\}$ be a family of $\ell$ unknown $k$-sparse vectors. Let $H_{\mathcal{B}} = (V, E_{\mathcal{B}})$ denote the $(\ell, k)$-sparse hypergraph induced by $\mathcal{B}$, i.e., the set of edges $E_{\mathcal{B}}$ consists of the supports of vectors from $\mathcal{B}$. Set $\alpha = 16\tau\ell$. By Lemma 7, there exists an $(\ell, k, \alpha)$-robust-resolvable matrix $X$ of size $m \times n$ with $m = \frac{(2k\ell-1)\log n}{w(\alpha, \ell, k)}(1 + o(1))$. By Definition 1, $H_{\mathcal{B}}$ can be uniquely recovered based on any noisy version $\tilde{z} \in \mathbb{Z}^m$ of the vector $z(X, H_{\mathcal{B}})$ so that $d_H(\tilde{z}, z(X, H_{\mathcal{B}})) < \alpha m$. Applying Lemma 2, with overwhelming probability one can learn some $\tilde{z}$ by asking $q = \mathcal{O}(\ell^2 m \log(nm))$ queries $\boldsymbol{x}^{(1)}, \ldots, \boldsymbol{x}^{(q)}$ and obtaining responses $y_1, \ldots, y_q$ provided that (1)-(2). Finally, to reconstruct the supports of all unknown vectors, we apply Algorithm 1. This algorithm uses a function $\mathbf{next}(\cdot)$ that takes an $(\ell, k)$-sparse hypergraph and outputs the next $(\ell, k)$-sparse hypergraph (here we assume some canonical order on the set of sparse hypergraphs). In other words, the algorithm involves enumerating over a family of $(\ell, k)$-sparse hypergraphs. Thereby, $q$ queries are sufficient for solving Problem 1.

---

**Algorithm 1:** Support recovery algorithm

**Data:** $(\ell, k, \alpha)$-robust-resolvable matrix $X$ of size $m \times n$ ;        /* exists by Lemma 7 */
      noisy response vector $\tilde{z}$ with $d_H(\tilde{z}, z(X, H_{\mathcal{B}})) \leq \alpha m$ ;        /* found by Lemma 2 */
**Result:** supports of the $\ell$ unknown vectors, $S_1, \ldots, S_\ell \subseteq [n]$
$H \leftarrow (V, E)$, where $V = [n]$, $E = \{e_1, \ldots, e_\ell\}$, $e_i = [k]$;
**while** $d_H(z(X, H), \tilde{z}) > \alpha m$ **do**
  | $H \leftarrow \mathbf{next}(H)$ ;                    /* outputs the next sparse hypergraph */
**end**
**for** $i \leftarrow 1$ *to* $\ell$ **do**
  | $S_i \leftarrow e_i$
**end**

---

## 5 Approximate recovery

Throughout this section, we assume that there is no noise in measurements, i.e., $\tau = 0$. Theorem 1 enables us to assume that the supports of all sparse vectors are already known. Thus, it remains to approximately reconstruct the values of the vectors.

A query vector $\boldsymbol{x} = (x_1, \ldots, x_n) \in \mathbb{R}^n$ is called *Gaussian* if each entry is sampled independently from a Gaussian distribution $\mathcal{N}(0, 1)$. We shall use the following result which says that Gaussian queries allow to efficiently reconstruct any sparse vector.

**Lemma 8** (Approximate vector recovery, [18]). *Let $\boldsymbol{\beta}$ be an unknown vector in $\mathbb{R}^k$ with $\|\boldsymbol{\beta}\|_2 = 1$. For any $\epsilon > 0$, there exists an algorithm that takes responses to Gaussian queries $\boldsymbol{x}^{(1)}, \ldots, \boldsymbol{x}^{(q)} \in \mathbb{R}^k$ with $q = \mathcal{O}(\frac{k}{\epsilon} \log \frac{k}{\epsilon})$ as input and outputs $\tilde{\boldsymbol{\beta}} \in \mathbb{R}^k$ so that $\left\| \boldsymbol{\beta} - \tilde{\boldsymbol{\beta}} \right\|_2 \le \epsilon$.*

Recall the response to a query in our model is taken uniformly at random from a set of $\ell$ possible responses. Therefore, we need to show how to model a response to a Gaussian query for any given vector from the family. Here, we will use our structural assumption regarding the support of vectors (c.f. Assumption 1). In Sec. 5.1, we show how to emulate individual Gaussian queries in our model. In Sec. 5.2 we present a two-stage approximate recovery algorithm.

### 5.1 Modeling Gaussian query

Define Inf to be a very large positive real number whose lower bound will be clear from the context. For a query vector $\boldsymbol{x} = (x_1, \ldots, x_n) \in \mathbb{R}^n$ and an integer $i \in [\ell]$, we write $\mathrm{Inf}^{(i)}(\boldsymbol{x})$ to denote the vector whose $j$-th entry is

$$\mathrm{Inf}^{(i)}(\boldsymbol{x})_j = \begin{cases} x_j, & j \in \mathrm{supp}(\boldsymbol{\beta}^{(i)}) \text{ or } j \notin \cup_{t \ne i} \mathrm{supp}(\boldsymbol{\beta}^{(t)}), \\ \mathrm{Inf}, & \text{otherwise.} \end{cases}$$

Let $\boldsymbol{0}$ denote an all-zero vector whose length will be clear from the context and $\boldsymbol{g} \in \mathbb{R}^n$ be a Gaussian query. Fix an integer $i \in [\ell]$ and some positive integer $m$. Set $\boldsymbol{x}^{(1,i)} = \ldots = \boldsymbol{x}^{(m/2,i)} = \mathrm{Inf}^{(i)}(\boldsymbol{g})$ and $\boldsymbol{x}^{(m/2+1,i)} = \ldots = \boldsymbol{x}^{(m,i)} = \mathrm{Inf}^{(i)}(\boldsymbol{0})$. Let $y_1, \ldots, y_m$ be responses to queries $\boldsymbol{x}^{(1,i)}, \ldots, \boldsymbol{x}^{(m,i)}$, i.e., $y_j = \mathrm{sign}(\langle \boldsymbol{x}^{(j,i)}, \boldsymbol{\beta} \rangle)$ with $\boldsymbol{\beta}$ being sampled uniformly at random from $\mathcal{B}$. Define a sample statistic that estimates the value $\mathrm{sign}(\langle \boldsymbol{g}, \boldsymbol{\beta}^{(i)} \rangle)$

$$\widetilde{\mathrm{sign}}(\boldsymbol{g}, \boldsymbol{\beta}^{(i)}, m) := \mathrm{round}\left( \frac{2\ell}{m} \left( \sum_{j=1}^{m/2} y_j - \sum_{j=m/2+1}^{m} y_j \right) + 1 \right).$$

**Lemma 9** (Gaussian query estimator). *Let $\mathcal{B}$ be a family of sparse vectors that satisfies Assumption 1. Then it holds that*

$$\Pr\left\{ \widetilde{\mathrm{sign}}(\boldsymbol{g}, \boldsymbol{\beta}^{(i)}, m) = \mathrm{sign}(\langle \boldsymbol{g}, \boldsymbol{\beta}^{(i)} \rangle) \right\} = 1 - \exp(-\Omega(\ell^{-2}m)).$$

### 5.2 Proof of Theorem 2

Let $\mathcal{B} = \{\boldsymbol{\beta}^{(1)}, \ldots, \boldsymbol{\beta}^{(\ell)}\}$ be a family of $\ell$ unknown $k$-sparse vectors that satisfy Assumption 1. Fix $\epsilon > 0$. First, we invoke Theorem 1 and with probability $1 - \mathcal{O}(1/n)$, the supports of all vectors in $\mathcal{B}$ can be reconstructed using $\mathcal{O}(k\ell^3 \max(2e\ell, 2ek)^{\min(\ell,k)} \log^2 n)$ queries. Note that the number of coordinates in the supports of the vectors is at most $k\ell$.

At the second stage, we first generate Gaussian vectors $g^{(1)}, \ldots, g^{(R)} \in \mathbb{R}^{k\ell}$ with $R = \mathcal{O}(\frac{k}{\epsilon} \log \frac{k}{\epsilon})$ being as in Lemma 8. Applying Lemma 9, with probability $1 - \mathcal{O}((n\ell\frac{k}{\epsilon} \log \frac{k}{\epsilon})^{-1})$ we can learn the value of $\mathrm{sign}(\langle g^{(j)}, \boldsymbol{\beta}^{(i)} \rangle)$ for some $j \in [R]$ and $i \in [\ell]$, by making $\mathcal{O}(\ell^2 \log(nk\ell/\epsilon))$ queries. By the union bound, all values $\{\mathrm{sign}(\langle g^{(j)}, \boldsymbol{\beta}^{(i)} \rangle) : j \in [R], i \in [\ell]\}$ are reconstructed with probability $1 - \mathcal{O}(1/n)$ by making $\mathcal{O}(\ell^3 \frac{k}{\epsilon} \log \frac{k}{\epsilon} \log(nk\ell/\epsilon))$ measurements. Finally, Lemma 8 guarantees that all sparse vectors from $\mathcal{B}$ can be recovered with the given precision $\epsilon$ based on the set of values $\{\mathrm{sign}(\langle g^{(j)}, \boldsymbol{\beta}^{(i)} \rangle) : j \in [R], i \in [\ell]\}$. Since $k\ell = \mathcal{O}(n)$, the required claim follows.

## 5.3 Proof of Theorem 3

Let $\mathcal{B} = \{\boldsymbol{\beta}^{(1)}, \ldots, \boldsymbol{\beta}^{(\ell)}\}$ be a family of $\ell$ unknown $k$-sparse vectors that satisfy Assumption 1. Fix $\epsilon > 0$. First, we apply Theorem 1 and with probability $1 - \mathcal{O}(1/n)$, the supports of all vectors in $\mathcal{B}$ can be reconstructed using $\mathcal{O}(k\ell^3 \max(2e\ell, 2ek)^{\min(\ell,k)} \log^2 n)$ queries. Note that the number of coordinates in the supports of the vectors is at most $k\ell$.

It remains to approximately recover the entries of the unknown vectors by asking queries in parallel to the queries used for the support recovery. Let $X$ be an $(\ell, k\ell)$-cover-free matrix of size $m \times n$ with $m = \mathcal{O}(k\ell \max(\ell, k)^{\min(k,\ell)} \exp(\min(\ell, k)) \log n)$ which exists by Lemma 6. Let $R = \mathcal{O}(\frac{k}{\epsilon} \log \frac{k}{\epsilon})$ be the number of queries required in Lemma 8. For $j \in [R]$, define $G^{(j)}(X)$ to be a real-valued matrix obtained from the binary matrix $X$ by replacing 1's by Inf's and 0's by independent samples from a Gaussian distribution $\mathcal{N}(0, 1)$. We define $\text{Inf}^{(j)}(X)$ to be a real-valued matrix obtained from the binary matrix $X$ by replacing 1's by Inf's.

We ask queries corresponding to rows of matrices $\{\text{Inf}^{(j)}(X), G^{(j)}(X) : j \in [R]\}$ multiple times. Let $T$ denote the number of times a fixed query is repeatedly queried. We set $T = \Theta(\ell^2 \log(n\ell R))$. Recall that all the matrices are obtained from an $(\ell, k\ell)$-cover-free matrix. This means that for every $j \in [R]$ and $i \in [\ell]$, there exist rows in $G^j(X)$ and $\text{Inf}^j(X)$ that are $\text{Inf}^{(i)}(\boldsymbol{g})$ and $\text{Inf}^{(i)}(\mathbf{0})$ for some Gaussian query $\boldsymbol{g}$. Thus, by Lemma 9, with probability $1 - \mathcal{O}((n\ell R)^{-1})$ one can learn $\text{sign}(\langle \boldsymbol{g}^{(j,i)}, \boldsymbol{\beta}^{(i)} \rangle)$ for some $i \in [\ell]$ and some independent Gaussian query $\boldsymbol{g}^{(j,i)}$, $j \in [R]$. By the union bound, one can learn the set of values $\{\text{sign}(\langle \boldsymbol{g}^{(j,i)}, \boldsymbol{\beta}^{(i)} \rangle) : j \in [R], i \in [\ell]\}$ with probability $1 - \mathcal{O}(1/n)$. Finally, Lemma 8 guarantees that all sparse vectors from $\mathcal{B}$ can be reconstructed with the given precision $\epsilon$ based on the set of values $\{\text{sign}(\langle \boldsymbol{g}^{(j,i)}, \boldsymbol{\beta}^{(i)} \rangle) : j \in [R], i \in [\ell]\}$.

Finally, we estimate the number of queries used at the second stage of the approximate recovery algorithm as $\mathcal{O}(\frac{k^2}{\epsilon} \log(\frac{k}{\epsilon}) \ell^3 \max(\ell, k)^{\min(k,\ell)} \exp(\min(\ell, k)) \log n \log \frac{n}{\epsilon})$. Thus, the total number of queries sufficient for solving Problem 2 can be estimated as $\mathcal{O}(\frac{k^2}{\epsilon} \ell^3 \max(2e\ell, 2ek)^{\min(k,\ell)} \log n \log(\frac{n}{\epsilon}) \log(\frac{k}{\epsilon}))$.

**Remark 3.** *We note the following improvement on query complexity if the number of queries required in Lemma 8 can be additionally bounded as $o(\log n)$ (e.g., it is applicable when $\epsilon = \Omega((\log n)^{-1/2})$). In such a case, one can generate $(\ell, k\ell, \alpha)$-robust-cover-free matrix of size $m \times n$ with small positive $\alpha$ and $m = \mathcal{O}(k\ell \max(\ell, k)^{\min(k,\ell)} \exp(\min(\ell, k)) \log n)$ and construct only two matrices $G(X)$ and $\text{Inf}(X)$ in a similar way. By performing the rest of the same procedure, one can decrease the total number of required queries in $\Omega(\frac{k}{\epsilon} \log(\frac{k}{\epsilon}))$ times.*

# 6 Conclusion

In this paper, we studied two problems that arise in recovering unknown sparse vectors from a mixture of sign responses. For the support recovery problem, we found a completely non-adaptive solution that does not use any assumptions. This problem was posed in [14]. In addition, the proposed algorithm is resilient to noisy measurements. When describing our result, we introduced the notion of a robust resolvable matrix and derived an existential result for such matrices. We leave the problem of deriving fundamental limits of this combinatorial object as an open challenging problem that can be of independent interest in computer science, coding theory, and combinatorics. For the approximate recovery problem, we suggested single-stage and two-stage reconstruction algorithms working under Assumption 1. Finding an algorithm for robust approximate recovery of a family of unknown sparse vectors without any assumptions remains a major open problem.

## Acknowledgments and Disclosure of Funding

The work was conducted in part when Nikita Polyanskii was with the Technical University of Munich and the Skolkovo Institute of Science and Technology. The author is indebted to Arya Mazumdar for bringing this problem to his attention. The research was supported by the German Research Foundation (Deutsche Forschungsgemeinschaft, DFG) under Grant No. WA3907/1-1 and by the Russian Foundation for Basic Research under Grant No. 20-01-00559.

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
