# A    Supplementary Material

## A.1    Proof of Lemma 1

Before proceeding with a proof of this statement, we recall the famous Hoefding inequality [17] which can be applied to bounded independent random variables, regardless of their distribution.

**Proposition 1** (Hoefding's inequality for bounded variables)**.** *Let $\xi_1, \ldots, \xi_n$ be independent random variables such that $a \leq \xi_i \leq b$ for all $i \in [n]$. Set $\xi = \sum \xi_i$ and $\mu = \mathbf{E}[\xi]$. Then, for all $\delta > 0$*

$$\Pr\left\{|\xi - \mu| \leq n\delta\right\} \geq 1 - 2\exp(-2\delta^2 n/(b-a)^2).$$

*Proof.* Let $Q \subseteq [n]$. Let $\boldsymbol{x} = (x_1, \ldots, x_n)$ be a vector such that $x_i$ has distribution $U(0,1)$ for $i \in Q$ and $x_i = 0$ otherwise. Define the random variable $\theta := \text{sign}(\boldsymbol{x}, \boldsymbol{\beta}_1) - \text{sign}(-\boldsymbol{x}, \boldsymbol{\beta}_2)$, where $\boldsymbol{\beta}_1$ and $\boldsymbol{\beta}_2$ are i.i.d. random variables taken uniformly at random from the set of unknown vectors $\mathcal{B}$. Clearly, $\mathbf{E}[\theta] = \frac{2}{\ell}\text{pir}(Q, \mathcal{B})$. From the lemma condition, it follows that there exists a set $I \subseteq [m]$ of size at least $(1 - \frac{1}{16\ell})m$ such that $y_i = \text{sign}(\langle \boldsymbol{x}^{(i)}, \boldsymbol{\beta} \rangle)$ for all $i \in I$. Moreover, one can find a subset $I' \subseteq I$ such that $|\{i \in I' : i \leq m/2\}| = |\{i \in I' : i > m/2\}| = \frac{8\ell-1}{16\ell}m$. By applying Proposition 1 for a set $\{y_i : i \in I', i \leq m/2\} \cup \{y_i : i \in I', i > m/2\}$ with $\xi := \sum_{i \in I'} y_i$, $\mu = \mathbf{E}[\xi] = \frac{|I'|}{2}\frac{2}{\ell}\text{pir}(Q, \mathcal{B}) = \frac{(8\ell-1)m}{8\ell^2}\text{pir}(Q, \mathcal{B})$, $a = -1$, $b = 1$ and $\delta = 1/(2\ell)$, we get

$$\Pr\left\{|\xi - \mu| \leq n\delta\right\} \geq 1 - 2\exp\left(-\Omega(m\ell^{-2})\right).$$

Thus,

$$\Pr\left\{\widetilde{\text{pir}}(Q, m) = \text{pir}(Q, \mathcal{B})\right\} =$$
$$= \Pr\left\{\text{pir}(Q, \mathcal{B}) - \frac{1}{2} < \frac{\ell}{m}\left(\sum_{i=1}^{m/2} y_i - \sum_{i=m/2+1}^{m} y_i\right) < \text{pir}(Q, \mathcal{B}) + \frac{1}{2}\right\}$$
$$\geq \Pr\left\{|\xi - \mu| \leq n\delta\right\}$$
$$\geq 1 - 2\exp\left(-\Omega(m\ell^{-2})\right).$$

This yields that $\widetilde{\text{pir}}(Q, m)$ is a correct estimate of $\text{pir}(Q, \mathcal{B})$ with probability $1 - \exp(-\Omega(m\ell^{-2}))$.
$\square$

## A.2    Proof of Lemma 2

*Proof.* By Lemma 1, one value $z(Q_i, H_{\mathcal{B}})$ can be correctly estimated with probability $1 - 1/(nm)$ by asking $q'$, $q' = O(\ell^2 \log(nm))$, queries provided that a fraction of the erroneous responses to these queries is at most $1/(16\ell)$. We ask $q = mq'$ queries and try to estimate all values $\{z(Q_i, H_{\mathcal{B}}), i \in [m]\}$. Recall that the total number of erroneous responses is bounded by $\tau q$. By the union bound, with probability $1 - 1/n$, the number of incorrect estimates is at most $\frac{\tau q}{q'/(16\ell)} = 16\ell\tau m$. This completes the proof.
$\square$

## A.3    Proof of Lemma 6

*Proof.* We shall make use of the random coding with expurgation technique. Let $r := \alpha m$. Consider a random binary matrix $X$ of size $m \times n$ whose entries are i.i.d. Bernoulli random variables which take the value 1 with probability $p = \frac{c}{c+f}$ and the value 0 with probability $1 - p$. Let $\mathcal{E}_t$ denote the event that the $t$-th column of $X$ is *bad*, i.e., there exist some disjoint subsets $U \subseteq [n]$, $t \in U$, $|U| = c$, and $W \subseteq [n]$, $|W| = f$, and less than $2r + 1$ rows $i \in [m]$ such that $X_{i,j} = 1$ for all $j \in U$ and $X_{i,j} = 0$ for all $j \in W$. Let $\nu$ be a Binomial random variable with parameters $m$ and $\overline{p} := p^c(1-p)^f$. It is clear that $\nu$ describes the number of rows in $X$ that separate two sets of columns

$U$ and $W$. By the union bound, we obtain

$$\Pr\{\mathcal{E}_t\} \leq \sum_{\substack{U,W \subseteq [n],\ t \in U \\ |U|=c, |W|=f,\ U \cap W = \emptyset}} \Pr\{\nu \leq 2r\}$$

$$\leq \binom{n-1}{c+f-1}\binom{c+f-1}{f}\sum_{i=0}^{2r}\binom{m}{i}\overline{p}^i(1-\overline{p})^{m-i}.$$

Note that the property on $\alpha$ in the statement of this lemma implies that $2r \leq \overline{p}m$. Applying the monotonicity property of the Binomial distribution, we obtain

$$\Pr\{\mathcal{E}_t\} \leq (2r+1)n^{c+f-1}\binom{m}{2r}(1-\overline{p})^{m-2r}\overline{p}^{2r}.$$

If the above probability is bounded above by $1/2$, then $X$ contains on average at most $n/2$ bad columns. This would imply the existence of an $(c, f, \alpha)$-robust-cover-free matrix of size $m \times n/2$. In order to have $\Pr\{\mathcal{E}_t\} \leq 1/2$, it suffices to take $m$ satisfying

$$(c+f-1)\log n + mH_2(2\alpha) + m(1-2\alpha)\log(1-\overline{p}) + 2\alpha m \log \overline{p} \leq o(m).$$

Thus, a required matrix exists if it holds that

$$m \geq \frac{(c+f-1)\log n}{(2\alpha-1)\log(1-\overline{p}) - 2\alpha \log \overline{p} - H_2(2\alpha)}(1+o(1)).$$

$\square$

## A.4   Proof of Lemma 9

*Proof.* To make everything work, we indeed need to use some assumptions regarding the support of vectors and the magnitude of all the entries: (a) the support of any vector from $\mathcal{B}$ is not fully included to the support of any other vector from $\mathcal{B}$, (b) the absolute value of each non-zero entry of $\boldsymbol{\beta}^{(i)}$ is bounded below by $c_l$ and above by $c_u$.

Under Assumption 1, for $j \neq i$ and large enough Inf, it holds that

$$s_{ij} := \mathrm{sign}(\langle \mathrm{Inf}^{(i)}(\boldsymbol{g}), \boldsymbol{\beta}^{(j)}\rangle) = \mathrm{sign}(\langle \mathrm{Inf}^{(i)}(\boldsymbol{0}), \boldsymbol{\beta}^{(j)}\rangle),$$

since the contribution of entries indexed by elements of $\mathrm{supp}(\boldsymbol{\beta}^{(i)})$ to both inner products is negligible as $\mathrm{Inf} \to \infty$. We also note that

$$\mathrm{sign}(\langle \mathrm{Inf}^{(i)}(\boldsymbol{0}), \boldsymbol{\beta}^{(i)}\rangle) = \mathrm{sign}(\langle \boldsymbol{0}, \boldsymbol{\beta}^{(i)}\rangle) = 1,$$
$$\mathrm{sign}(\langle \mathrm{Inf}^{(i)}(\boldsymbol{g}), \boldsymbol{\beta}^{(i)}\rangle) = \mathrm{sign}(\langle \boldsymbol{g}, \boldsymbol{\beta}^{(i)}\rangle),$$

Set $\xi = \sum_{j=1}^{m/2} y_j - \sum_{j=m/2+1}^{m} y_j$. Then $\mu = \mathbf{E}[\xi] = \frac{m}{2\ell}(\mathrm{sign}(\langle \boldsymbol{g}, \boldsymbol{\beta}^{(i)}\rangle) - 1)$. By applying Proposition 1, we obtain

$$\Pr\left\{\mathrm{sign}(\langle \boldsymbol{g}, \boldsymbol{\beta}^{(i)}\rangle) - \frac{1}{2} < \frac{2\ell}{m}\xi + 1 < \mathrm{sign}(\langle \boldsymbol{g}, \boldsymbol{\beta}^{(i)}\rangle) + \frac{1}{2}\right\}$$

$$= \Pr\left\{\frac{m}{2\ell}\left(\mathrm{sign}(\langle \boldsymbol{g}, \boldsymbol{\beta}^{(i)}\rangle) - \frac{1}{2}\right) < \xi + \frac{m}{2\ell} < \frac{m}{2\ell}\left(\mathrm{sign}(\langle \boldsymbol{g}, \boldsymbol{\beta}^{(i)}\rangle) + \frac{1}{2}\right)\right\}$$

$$= \Pr\left\{|\xi - \mu| < \frac{m}{4\ell}\right\}$$

$$\geq 1 - 2\exp\left(-\Omega(\ell^{-2}m)\right).$$

This yields the required statement.

$\square$