# OpenReview forum: "On learning sparse vectors from mixture of responses"
_NeurIPS.cc/2021/Conference — NeurIPS 2021 Poster_

### Official Review · Reviewer_WBt8 · 2021-07-16

**Rating:** 4
**Confidence:** 2

**Summary:**

This paper studied two learning problems.

For the first problem, they aim at designing queries such that all sparse vectors from the family  can be reconstructed using a sequence of responses. Each response to a query  vector shows the sign of the inner product between the query vector and a random vector from the family.

For the second problem, they study robust learning using only the support of all vectors from the family in the setting in which a fraction of responses can be flipped.


**Limitations And Societal Impact:**

Limitations are not discussed. Societal impact is not directly discussed, but it's a theory paper, so it does not have a direct impact.

**Main Review:**

Strength: This paper studied two learning problems and provided some theoretical results.

Weakness :

1. The paper is poorly-organized and hence hard to follow. Due to the structure issue, it is not sure that the contribution is large enough.

2. Reference starts at page 9, not sure if this violates the guidelines of the conference.

3. Not sure the methods proposed in this paper are novel enough.

4. Several 1bit compressed sensing papers are missing:

“Learning and 1-bit Compressed Sensing under Asymmetric Noise’’

“One-Bit ExpanderSketch for One-Bit Compressed Sensing.’’

“On Fast Decoding of High-Dimensional Signals from One-Bit Measurements.’’

5. Several mixtures of linear regression papers are missing:

“Learning Mixtures of Linear Regressions with Nearly Optimal Complexity.’’

“Learning mixtures of linear regressions in subexponential time via fourier moments.’’

“Small covers for near-zero sets of polynomials and learning latent variable models.‘’


**Time Spent Reviewing:**

5

---

> ### Author Response · Authors · 2021-08-06
> **Response**
>
> Thank you for providing your review! Below please find our comments:
>
> 1Q: The paper is poorly-organized and hence hard to follow. Due to the structure issue, it is not sure that the contribution is large enough.
>
> 1A: If you find our paper hard to read, please write exactly what has to be changed. We would be glad to improve our presentation.
>
> 2Q: Reference starts at page 9, not sure if this violates the guidelines of the conference.
>
> 2A: We asked the organizers regarding the formatting instructions for this year. It is more than fine to have even less than 9 pages (say, 7 or 8) of the content. Anyway, we will incorporate several changes so that references would start on page 10.
>
> 3Q: Not sure the methods proposed in this paper are novel enough.
>
> 3A: Our basic ideas are not novel, but we originally combine them to solve an open problem from [GMP'20] and some new problem statements.
>
> 4-5Q: Several 1bit compressed sensing papers are missing. Several mixtures of linear regression papers are missing.
>
> 4-5A: We believe that the most relevant literature is cited in our paper. Nevertheless, we are glad that you draw our attention to these works. We will incorporate several changes to references.
>
> 6Q: Limitations are not discussed. Societal impact is not directly discussed, but it's a theory paper, so it does not have a direct impact.
>
> 6A: The limitations of the paper are mentioned in the concluding section and in the last paragraph of section 1.1 and Remark 1.

---

### Official Review · Reviewer_jx9h · 2021-07-17

**Rating:** 6
**Confidence:** 3

**Summary:**

The paper describes a method to recover a set of sparse vectors, either their values or their supports, by sampling them at random and obtaining their linear measurement quantized to a single bit representing the measurement sign. The crux of the analysis focuses on (i) modeling the set of sparse vectors as a hypergraph, where each vector support corresponds to an edge; (ii) defining a set of linear observations of signals that translates into observations of edges in the hypergraph; (iii) demonstrating that given enough noisy edge observations from the hypergraph and for sufficiently simple regimes (number and complexity of the edges, and noise level), the set of edges defining it can be estimated at sufficient accuracy.

**Limitations And Societal Impact:**

Given the flexibility of the proposed framework, limitations with respect to results in the literature are discussed (e.g., Remark 1). The paper also mentions the fact that the computational effort of recovery is not being considered, and that the measurement vectors are considered in terms of existence rather than construction.

**Main Review:**

There is significant sophistication in the analysis of the proposed method that makes the read a bit difficult, but the presentation is clear enough after a re-read. In particular, the connection between Lemma 1 and Lemma 2 is not sufficiently clear, but Lemma 2 motivates well the rest of the presentation.

In the results, it is not always clear how the noise level tau > 0 has an effect (e.g., Theorems 2 and 3). In fact, the differences between the two halves of Figure 5 are minor, even for the baselines.

To improve clarity, terms as "precision epsilon" should be defined; some minor typos (e.g., "to recovery all vectors"; h in the last row of Page 6) exist.

**Time Spent Reviewing:**

5 hours

---

> ### Author Response · Authors · 2021-08-05
> **Response**
>
> Thank you for your review and important notes! We write our responses below:
>
> 1A: There is significant sophistication in the analysis of the proposed method that makes the read a bit difficult, but the presentation is clear enough after a re-read. In particular, the connection between Lemma 1 and Lemma 2 is not sufficiently clear, but Lemma 2 motivates well the rest of the presentation.
>
> 1Q: Thank you for pointing out this issue. We are planning to improve the readability of Section 4 in the further update.
>
> 2A: In the results, it is not always clear how the noise level tau > 0 has an effect (e.g., Theorems 2 and 3). In fact, the differences between the two halves of Figure 5 are minor, even for the baselines.
>
> 2Q: For the $\epsilon$-approximate recovery problem, we assume that $\tau = 0$ (this is stated in Problem 1 and Section 1.1). Anyway we will clarify this point since it is quite important. In Theorem 1, $\tau$ changes only the constant factor in front of the main term. But, this constant can be rather large when $\tau$ is close to the border value. By the way, we didn't get a comment regarding Figure 5.
>
> 3A: To improve clarity, terms as "precision epsilon" should be defined; some minor typos (e.g., "to recovery all vectors"; h in the last row of Page 6) exist.
>
> 3Q: We will improve clarity in the next update. Thank you for noticing these inaccuracies.

---

> > ### Comment · Reviewer_jx9h · 2021-08-19
> > **Response**
> >
> > On 2Q, my mistake on the comment referring to a figure; it must have been a mixup between reviews.
> >
> > Thank you for the responses to the other comments.

---

### Official Review · Reviewer_xjw9 · 2021-07-19

**Rating:** 5
**Confidence:** 2

**Summary:**

The paper provides statistical upper bounds for the problem of learning a mixture of sparse vectors from signs of linear queries, both in the noiseless and noisy settings.

**Main Review:**

The paper studies the problem of proper learning of a mixture of sparse vectors from possibly noisy 1-bit quantized linear queries.

In the noiseless setting, the authors prove an information-theoretic upper bound on the query complexity that uses fewer technical assumptions than the main result of Gandikota-Mazumdar-Pal (Neurips '20). In the noisy setting, the authors prove a similar upper bound but for the weaker problem of support recovery under certain identifiability assumptions on the underlying mixture.

The paper introduces the use of a number of interesting coding theoretic principles (piercing numbers, robust resolvable binary matrices, robust cover-free codes) to a fairly challenging family of problems.

Despite its contributions, I have several concerns -- primarily with how the results of this paper are presented -- which prevent a higher score.
1. To complement their upper bounds, [GMP'20] give a concrete, easy-to-implement (ie., low degree polynomial-time) algorithm for the same problem, and also show numerical results on standard ML benchmark datasets. This paper does not provide any algorithmic upper bounds or experiments; indeed, the algorithm "seems to" involve enumerating over a family of sparse hypergraphs which can be exponentially large in either the sparsity or the number of components, or both. I doubt that such approaches can ever find their way to practical use cases such as recommender systems.
2. I put "seems to" in the above comment in quotes, since I can only make an educated guess of what the algorithm does. The actual learning procedure for either problem setting is never spelt out anywhere clearly in the paper. This might be a presentation issue but a rather serious one. I'd suggest that the authors transparently introduce the recovery algorithms in terms of pseudocode.
3. Remark 1 is a bit confusing (and in my view, misleading). The query complexity is exponentially worse than the bound in the noiseless case presented in [GMP'20], so I am not sure if it directly resolves the open problem posed in that paper.

---

I thank the authors for their responses. I am inclined to retain my score (but to be fair the authors, my above questions cannot easily be addressed in a rebuttal without a revision). Other reviewers seem to be viewing the paper positively, and I respect that. Overall, I would be fine with accepting the paper provided they clearly address the points above in their final revision.

**Time Spent Reviewing:**

8

---

> ### Author Response · Authors · 2021-08-05
> **Response**
>
> Thank you for your review and useful remarks! We put our responses below:
>
> 1Q: To complement their upper bounds, [GMP'20] give a concrete, easy-to-implement (i.e., low degree polynomial-time) algorithm for the same problem, and also show numerical results on standard ML benchmark datasets. This paper does not provide any algorithmic upper bounds or experiments; indeed, the algorithm "seems to" involve enumerating over a family of sparse hypergraphs which can be exponentially large in either the sparsity or the number of components, or both. I doubt that such approaches can ever find their way to practical use cases such as recommender systems.
>
> 1A: Thank you for pointing out this issue. Our proposed learning scheme is indeed not efficient. It implicitly involves enumerating over a family of hypergraphs. We will definitely write about the complexity of our algorithm in the next update. By the way, we have a sentence in the last paragraph of Section 1.1 which says that the problem of designing an efficient algorithm (in terms of running time) is not in the focus of our study.
>
> 2Q: I put "seems to" in the above comment in quotes, since I can only make an educated guess of what the algorithm does. The actual learning procedure for either problem setting is never spelt out anywhere clearly in the paper. This might be a presentation issue but a rather serious one. I'd suggest that the authors transparently introduce the recovery algorithms in terms of pseudocode.
>
> 2A: We totally agree that it is a nice idea to precisely write how the algorithm should work and so, we will include pseudocodes for our learning strategies. On the other hand, we note that the current message of the paper is to show that it is theoretically possible to find the supports of unknown sparse vectors without any assumptions.
>
> 3Q: Remark 1 is a bit confusing (and in my view, misleading). The query complexity is exponentially worse than the bound in the noiseless case presented in [GMP'20], so I am not sure if it directly resolves the open problem posed in that paper.
>
> 3A: [GMP'20] left the support recovery problem without any assumptions as an open problem. Specifically, it was not clear if it is possible to learn the supports of an arbitrary family of three or more sparse vectors using any algorithm (even with large query complexity and algorithmic complexity). So, resolving the support recovery problem without any assumptions was our initial goal and we think that this problem is fully resolved in our paper. At the same time, we admit that our presentation of this point is not perfect and we must clarify the details.
>
> 3A (additional notes on the "exponential" bound): We also note that there is a related well-researched problem of learning hidden sparse hypergraphs using non-adaptive edge-detecting queries, where the binary response to a query is positive if the query contains at least one hyperedge and negative otherwise. In this setting, to learn an $(\ell,k)$-sparse hypergraph without any assumptions (c.f. [B'13, DVMT'02]) it is necessary to make at least $\Omega\left(\frac{(\ell+k)\binom{\ell+k}{k}}{\log\binom{\ell+k}{k}}\log n\right)$ queries. This lower bound is also exponential in the parameters $\ell$ and $k$.
> Strong assumptions can make the problem much easier. For instance, assume that hyperedges in a hypergraph are pairwise disjoint. Then one can argue that using union-free families, it is possible to find a hypergraph by asking at most $O(k^2\ell^2 \log n)$ queries.
>
> [B'13] N.H. Bshouty. Exact Learning from Membership Queries. Some Techniques, Results
> and New Directions. ALT 2013. pp. 33–52. (2013).
>
> [DVMT'02] D’yachkov, P. Vilenkin, A. Macula., D. Torney. Families of finite sets in which
> no intersection of l sets is covered by the union of s others. J. Comb Theory Ser A.
> 99. pp. 195–218. (2002).

---

### Official Review · Reviewer_zJ8u · 2021-07-20

**Rating:** 7
**Confidence:** 3

**Summary:**

This paper considers the problem of reconstructing a mixture of (sparse) linear classifiers from (noisy) binary responses. For each query, a sparse vector is selected from the ensemble and the inner product of the query vector is returned (perhaps plus noise). At a first glance, it is surprising that it is even possible to recover the underlying vectors reliably. This was established by Gandikota, Mazumdar, and Pal in a NeurIPS 2020 paper. This submission builds on this work by removing a structural assumption. This is a theoretical contribution and there are no simulations.

**Limitations And Societal Impact:**

I did not have any concerns.

**Main Review:**

As summarized above, this paper considers the interesting problem of recovering the underlying sparse vectors from a mixture of linear classifiers. Each query returns the inner product with one classifier from the mixture plus noise. In the 2020 work, it was assumed that the support of each classifier vector is not contained in the union of the supports of the other vectors. This submission significantly relaxes this assumption to only require that no classifier vector's support is fully contained in the support of any other vector. On the technical side, it relies on combinatorial constructions, such as robust-resolvable matrices. I think that overall this is a nice theoretical contribution to NeurIPS, as it improves significantly on a 2020 NeurIPS paper, using some interesting new math.

Here are a few items that I think might have improved this submission:
-Graphical/cartoon illustration of the improvement in query complexity over the 2020 paper
-Some discussion of any known impossibility results (e.g., lower bounds), even if trivial at this point
-Numerical simulations to compare this method to prior work, esp. the 2020 paper. (Note that the 2020 paper includes both synthetic and real data.)


**Time Spent Reviewing:**

2

---

> ### Author Response · Authors · 2021-08-05
> **Response**
>
> Thank you for your suggestions! We address your comments below:
>
> 1Q: Graphical/cartoon illustration of the improvement in query complexity over the 2020 paper
>
> 1A: We agree that for some small $\ell$ and $k$, it is a good idea to visualize the improvements. So, we will add such an illustration.
>
> 2Q: Some discussion of any known impossibility results (e.g., lower bounds), even if trivial at this point
>
> 2A: This is a very good idea. However, the dependence on $\ell$ is highly unclear. For example, as shown in Remark 1, it suffices to have $O\left(\frac{\ell\log^2 n}{\log\ell}\right)$ measurements for $k=1$. However, we will add a trivial bound that comes from the 1-bit compressed sensing setting.
>
> 3Q: Numerical simulations to compare this method to prior work, esp. the 2020 paper. (Note that the 2020 paper includes both synthetic and real data.)
>
> 3Q: Again a good point. However, at this stage, the decoding complexity of the proposed learning scheme seems to be large. Anyway, we will work on it since for reasonably small parameters, say $n=200$, $\ell=2$ and $k=4$ it is possible to get simulation results.

---

### Official Review · Reviewer_uLNB · 2021-08-01

**Rating:** 8
**Confidence:** 5

**Summary:**

This paper answers a number of open questions in [12] and generalizes some of them significantly as well. The contributions of the paper are summarized as follows:

1) The authors provide a querying scheme that can recover the support of all unknown vectors without any structural assumption on the unknown vectors. This result is a big step in the direction of answering a major open question raised in [12].

2) The results for support recovery in this paper is also resilient to adversarial noise. The idea of using robust resolvable matrices for recovering the support under this condition is very interesting.

3) The authors also provide a new technique for emulating Gaussian queries in this problem which works under milder structural assumptions than stipulated by [12]. Hence, this paper also results in improved results for Approximate recovery of the unknown vectors.

**Limitations And Societal Impact:**

Yes

**Main Review:**

The paper provide significantly interesting results using a number of novel ideas. I believe that the paper is quite well-written but the exposition can be improved in some places. Below, I have mentioned a few questions/suggestions that should be resolved in the final version.

1) In Theorem 1, how do we know that there exist a positive root of f(\alpha,\ell,k)? It would be good to characterize \alpha_0 for different values of \ell,k and visualize them in a plot.

2) The proof of Proposition 1 needs to be described with more details. First, the expectation of the piercing number estimator need to be mentioned and secondly, show why the adversarial error+ error due to randomness does not cause an issue in computing the piercing number correctly.

3) In Lemma 6, alpha is used to denote the real number in [0,\alpha_0). Some other variable must be used.

4) It would also be good to compare these results with the ones in a recent paper (https://arxiv.org/pdf/2106.05951.pdf) on the same problem.

**Time Spent Reviewing:**

4-5

---

> ### Author Response · Authors · 2021-08-05
> **Response**
>
> Thank you for your review and helpful remarks! Below we depict our responses to your questions/suggestions:
>
> 1Q: In Theorem 1, how do we know that there exists a positive root of $f(\alpha,\ell,k)$? It would be good to characterize $\alpha_0$ for different values of $\ell$, $k$ and visualize them in a plot.
>
> 1A: This is a good remark. The first positive root is actually $\alpha=p'/2$. This can be seen by taking the derivative of the function $f$ and computing $f$ at that point or this fact can be viewed from the proof since that point is the concentration point of Binomial distribution. So, we will slightly polish the statement and add a plot.
>
> 2Q: The proof of Proposition 1 needs to be described in more detail. First, the expectation of the piercing number estimator needs to be mentioned, and secondly, show why the adversarial error+ error due to randomness does not cause an issue in computing the piercing number correctly.
>
> 2A: Sure, we will add more details to this proof. We totally agree that the current presentation is rather short.
>
> 3Q: In Lemma 6, alpha is used to denote the real number in $[0,\alpha_0)$. Some other variable must be used.
>
> 3A: First of all, the statement of Lemma 6 will be polished due to arguments similar to 1A. Also, we agree that \alpha is used here in two different senses. We will improve the presentation of Lemma 6.
>
> 4Q: It would also be good to compare these results with the ones in a recent paper (https://arxiv.org/pdf/2106.05951.pdf) on the same problem.
>
> 4A: This is an interesting highly relevant manuscript that appeared on arXiv after the submission so that we didn't have a chance to compare our results. We will definitely add a comparison between our results and the results from that paper.

---

> > ### Comment · Reviewer_uLNB · 2021-08-23
> > **Question**
> >
> > Yes, I agree that $p'/2$ is a positive root of the function $f(\alpha,\ell,k)$. However, I do not understand the necessity to write Theorem 1 in such a difficult manner. However, why do you say that $p'/2$ is the first positive root? Why will there not exist any other positive root very close to zero? Do we get a simpler statement if we use $f(0,\ell,k)$ in the sufficient number of adaptive queries?

---

> > > ### Author Response · Authors · 2021-08-23
> > > **Response**
> > >
> > > $p'/2$ is the first positive root because of the following arguments. Compute the derivative $\frac{\partial f}{\partial \alpha} = 2\log(1-p')-2\log p'+2\log\left(\frac{2\alpha}{1-2\alpha}\right).$ Then $
> > > \frac{\partial f}{\partial \alpha} = 0 \Longleftrightarrow \left(\frac{1-p'}{p'}\right) = \frac{1-2\alpha}{2\alpha} \Longleftrightarrow \alpha=p'/2. $
> > > Since $f(0,\ell,k)=-\log(1-p')>0$, $f(p'/2,\ell,k)=0$ and $\frac{\partial f}{\partial \alpha}<0$ for $\alpha\in(0,p'/2)$, we conclude that $f$ is decreasing for $\alpha\in(0,p'/2)$ and $f>0$ for $\alpha\in[0,p'/2)$.
> > >
> > > Indeed, using big $O$ notation, we can re-write the statement in terms of $f(0,\ell,k)$ if $\tau$ is upper bounded properly. We will simplify the statement to make it easy to read.

---

> > > > ### Comment · Reviewer_uLNB · 2021-08-24
> > > > **Satisfactory Response**
> > > >
> > > > I thank the authors for the nice explanation. I will keep my score of 8 since I do not have any other concerns. I also feel that numerical experiments are not necessary for this paper just to tick some checkbox and the theoretical results are enough to justify its score. Saying that, it would be good to spell out the actual algorithm clearly in the subsequent version. It would also be good if the computational complexity of the algorithm is clearly spelled out in the main theorem or in a subsequent remark.

---

### Decision · Program_Chairs · 2021-09-28

**Decision:**

Accept (Poster)

**Comment:**

Based on the reviews and subsequent discussion, it is clear that this paper is tackling a non-trivial problem and contains good results. However there are some concerns with clarity and writing, and we ask the authors to incorporate the suggestions given by the reviewers. There is also a suggestion to make it clear in the abstract and in the beginning of the introduction that the problems considered were actually studied before, and are not introduced by the authors.

**Consistency Experiment:**

NeurIPS has a long history of experimentation. In 2014, NeurIPS ran an experiment in which 10% of submissions were reviewed by two independent committees to quantify the randomness in the review process. This year, we repeated a variant of this experiment to see how the quality of the review process has changed over time.  This paper was part of the experiment and was therefore assigned to two committees (consisting of reviewers, an Area Chair, and a Senior Area Chair) that reached independent decisions.  If both committees made the same recommendation, this recommendation was followed. If a single committee recommended acceptance, the paper was accepted (with the exception of a few cases in which the other committee identified what we considered a fatal flaw, e.g., an error in a key result).

This copy’s committee reached the following decision: **Accept (Poster)**

The other committee assigned to the paper recommended **Reject**.  You can find the other set of reviews, along with any follow up discussion with the authors here:
https://openreview.net/forum?id=XOAjcE5GerI